# Introducing Electrode Contact by Controlled Micro-Alloying in Few-Layered GaTe Field Effect Transistors

**Xiuxin Xia [1,2,†], Xingdan Sun [1,2,†], Hanwen Wang [1,2,*] and Xiaoxi Li [1,2,3,*]**

1   Shenyang National Laboratory for Materials Science, Institute of Metal Research, Chinese Academy of Sciences, Shenyang 110016, China; xxxia17s@imr.ac.cn (X.X.); xdsun15s@imr.ac.cn (X.S.)
2   School of Material Science and Engineering, University of Science and Technology of China, Hefei 230026, China
3   Collaborative Innovation Center of Extreme Optics, Shanxi University, Taiyuan 030006, China
*   Correspondence: hwwang15s@imr.ac.cn (H.W.); xxli@imr.ac.cn (X.L.)
†   These authors contributed equally to this work.

**Abstract:** Recently, gallium telluride (GaTe) has triggered much attention for its unique properties and offers excellent opportunities for nanoelectronics. Yet it is a challenge to bridge the semiconducting few-layered GaTe crystals with metallic electrodes for device applications. Here, we report a method on fabricating electrode contacts to few-layered GaTe field effect transistors (FETs) by controlled micro-alloying. The devices show linear *I-V* curves and on/off ratio of $\sim 10^4$ on $HfO_2$ substrates. Kelvin probe force microscope (KPFM) and energy dispersion spectrum (EDS) are performed to characterize the electrode contacts, suggesting that the lowered Schottky barrier by the diffusion of Pd element into the GaTe conduction channel may play an important role. Our findings provide a strategy for the engineering of electrode contact for future device applications based on 2DLMs.

**Keywords:** contact; alloying; GaTe; Pd electrode

## 1. Introduction

Due to the versatile properties of two-dimensional layered materials (2DLMs) [1–3], novel phenomena have been demonstrated recently, such as spintronics [4–6], optoelectronics [7,8], ferromagnetism [9], and superconductor-insulator transition [10]. As a newly emerging two-dimensional layered material with low lattice symmetry and a direct bandgap of $\sim 1.67$ eV, gallium telluride (GaTe) offers excellent opportunities for nanoelectronic devices such as photodetectors [11], anisotropic floating-gate memories [12], and radiation detectors [13]. Moreover, thickness-induced structural phase transformation [14], solution-processed GaTe [15], and bandgap restructuring [16] were also reported in this material. Therefore, this makes GaTe a candidate for future electronic applications.

As is well known, due to the work function mismatch, the Schottky barrier [17] formed at the interfaces between metal and semiconductors significantly limits the injection of electron and hole into the conduction channel. Meanwhile, high contact resistance masks the nature of material itself, especially at low temperature. Generally, analogous to many other two-dimensional semiconductors, the performance of GaTe devices is severely limited by electrical contact problems [18,19]: the on/off ratios are often relatively low ($\sim 10^2$) and the *I-V* curves are often nonlinear. Therefore, to obtain GaTe devices with high-performance, reducing or even eliminating the Schottky barrier at the interfaces between metal and semiconductor is of vital importance. Achieving barrier-free contact

with linear current-voltage output characteristics remains a significant challenge for the applications of two-dimensional semiconductor devices.

Over the past several years, tremendous efforts have been carried out to optimise the contact resistance for 2D nanoelectronics [20–29] such as using work function-matched metals [26], phase engineering [24,30], van der Waals heterostructure [21,31], selective etching [23,32], tunnelling effect [20,25,28], etc. The motivation of polymorph engineering or van der Waals contact is to decrease the contact barrier or increase carrier mobility, yet these methods always need sophisticated manipulation skills, and an easy methodology to tackle this problem is highly desired. Alloying is another method for higher carrier density causing the regrown alloyed district at the contact area to be highly doped, which has an impact on contact resistance and transport properties consequently [33]. Except for GaAs, alloyed ohmic contact has been reported in other semiconductor systems such as InP [34] for ohmic contact with control of heating time. Hence, refining the contact impedance by alloying via rapid thermal annealing provides more possibilities for simple approaches.

Here, we propose a micro-alloying method by employing Pd/Ti/Au electrodes to contact the few-layered GaTe. We carried out Kelvin probe force microscope (KPFM) and electron disperse spectroscopy (EDS) analysis for further understanding this type of micro-alloying contact.

## 2. Materials and Methods

As shown in Figure 1a, GaTe is a layered material which has a monoclinic structure with the C2/m space-group symmetry. In this paper, GaTe single crystal was synthesised by self-flux method. Ingredient powders with high purity (99.99%) were mixed with stoichiometric ratio of Ga:Te = 1:1, the mixture was kept at 880 °C for 3 h and then was cooled to 700 °C at a rate of 1.5 °C per minute followed by a natural cooling process. The quality of as-grown crystal was confirmed by X-ray diffraction and energy dispersion spectrum. As can be seen in Figure 1b, X-ray diffraction (XRD) result indicates high quality single crystal was obtained. For the GaTe FETs fabrication, GaTe flakes were mechanically exfoliated onto a $SiO_2$/Si substrate coated with 10 nm $HfO_2$ in a glove box to avoid any possible degradation induced by water or oxygen, and atomic force microscopy (AFM) was used to determine their thickness and roughness. High-k $HfO_2$ was grown on the surface of $SiO_2$/Si by atomic layer deposition (ALD), prior to the exfoliation of GaTe crystal. Metal electrodes, including Ti/Au (5/30 nm) and Pd/Ti/Au (10/5/20 nm), were fabricated by standard electron beam lithography (EBL) and electron beam evaporation (EBV), palladium, titanium, and gold were deposited in sequence after the patterning of the devices. After lift-off, the devices were annealed in a chemical vapor deposition furnace in argon and hydrogen atmosphere (20:2) at 320 °C. We performed electrical characterisations on our devices under room temperature using a semiconductor analyzer (Agilent B1500A, Agilent Technologies, Inc., Santa Clara, SA, USA) and a probe station (Cascade EPS150, Cascade Microtech Inc., Beaverton, SA, USA).

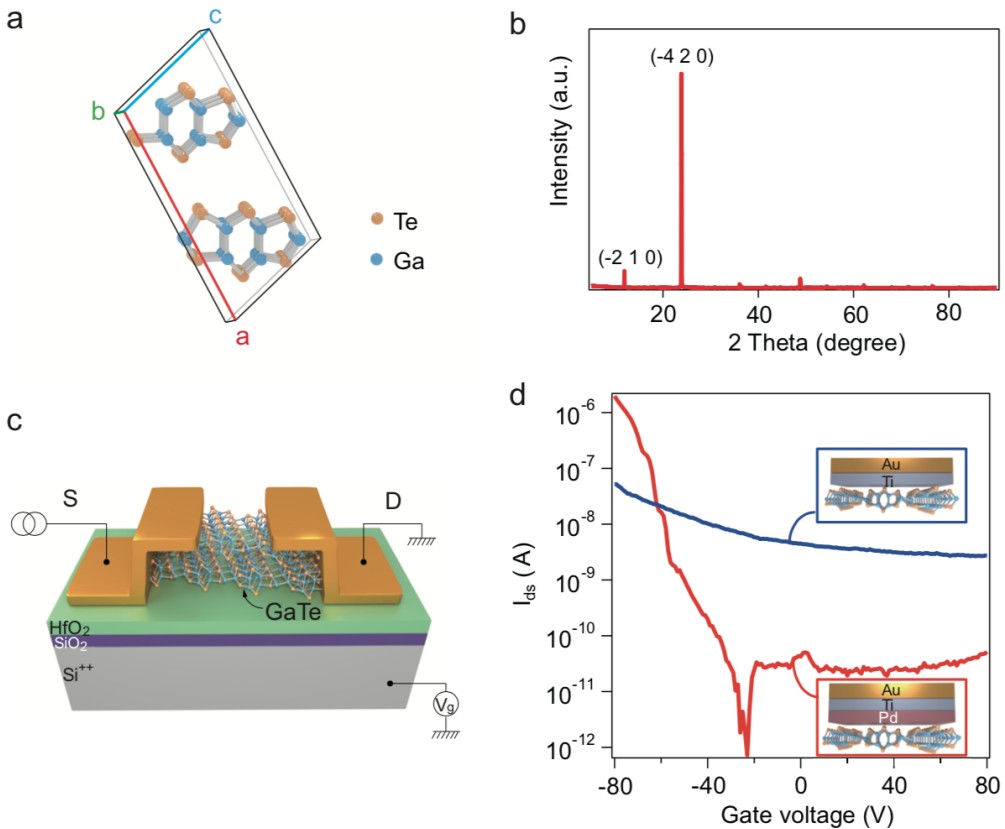

**Figure 1.** (**a**) Schematic of a unit cell of the GaTe crystal. The interlayer spacing is 0.8 nm. (**b**) XRD pattern of bulk GaTe grown by self-flux method. Crystallographic index of (−2 1 0) and (−4 2 0) are marked. (**c**) Schematic of device with Pd/Ti/Au electrodes on HfO$_2$. (**d**) The log scale of typical transfer curves of few-layered GaTe field effect transistors with Ti/Au and Pd/Ti/Au electrodes, respectively. It is seen that alloyed Pd/Ti/Au electrodes yield higher on-state current and much improved on/off ratio.

## 3. Results and Discussion

First, we characterised the few-layered GaTe FET on HfO$_2$ substrates with Ti/Au electrodes. As shown by the blue solid curve in Figure 1d, the performance of the device is poor in general, the device cannot be turned off even at high gate doping (such as $V_g$ = +80 V), and the current of the device can only reach a few tens of nA. As for the device with Pd/Ti/Au electrodes with the same preparation process and testing condition, the performance of the device has changed dramatically. As shown by the red solid curve in Figure 1d, compared to the Ti/Au electrodes, a noticeably higher current of μA in the GaTe device with Pd/Ti/Au electrode is observed. Moreover, higher on/off ratio of ∼10$^4$ is also achieved in the devices with Pd/Ti/Au electrodes. Similar behaviour was seen in multiple devices with various thicknesses of GaTe flakes.

Based on the enhanced performance of GaTe devices with Pd/Ti/Au (as compared with those using Ti/Au electrodes), we illustrate the simplified energy band diagram of GaTe, Pd, and Ti, respectively, as shown in Figure 2a. Obviously, a lower interfacial contact barrier can be seen between GaTe and Pd. The work function of Pd is 5.22–5.60 eV, which is very close to that of GaTe (5.14 eV). Furthermore, interestingly, we found that the Pd element in the Pd/Ti/Au electrodes of the GaTe FETs can diffuse into the interface between the GaTe electrodes and the adjacent areas with the annealing process, we call it a micro-alloying process that may contribute to improving the work function match at the electrode/GaTe interface. Detailed discussions will be available in the subsequent parts of this paper. Here a quasi-linear dependence can be seen between the annealing time and the diffusion distance (Figure 2b), which reaches up to 2 μm after 1 h annealing. We also demonstrate the field effect

curves with different annealing time in devices with alloyed Pd/Ti/Au electrodes, shown in Figure 2c. Conspicuously, significantly improved electrical performance of the device has been obtained with longer annealing time. The on-state current reached up to 2 μA after 60 min annealing, for example.

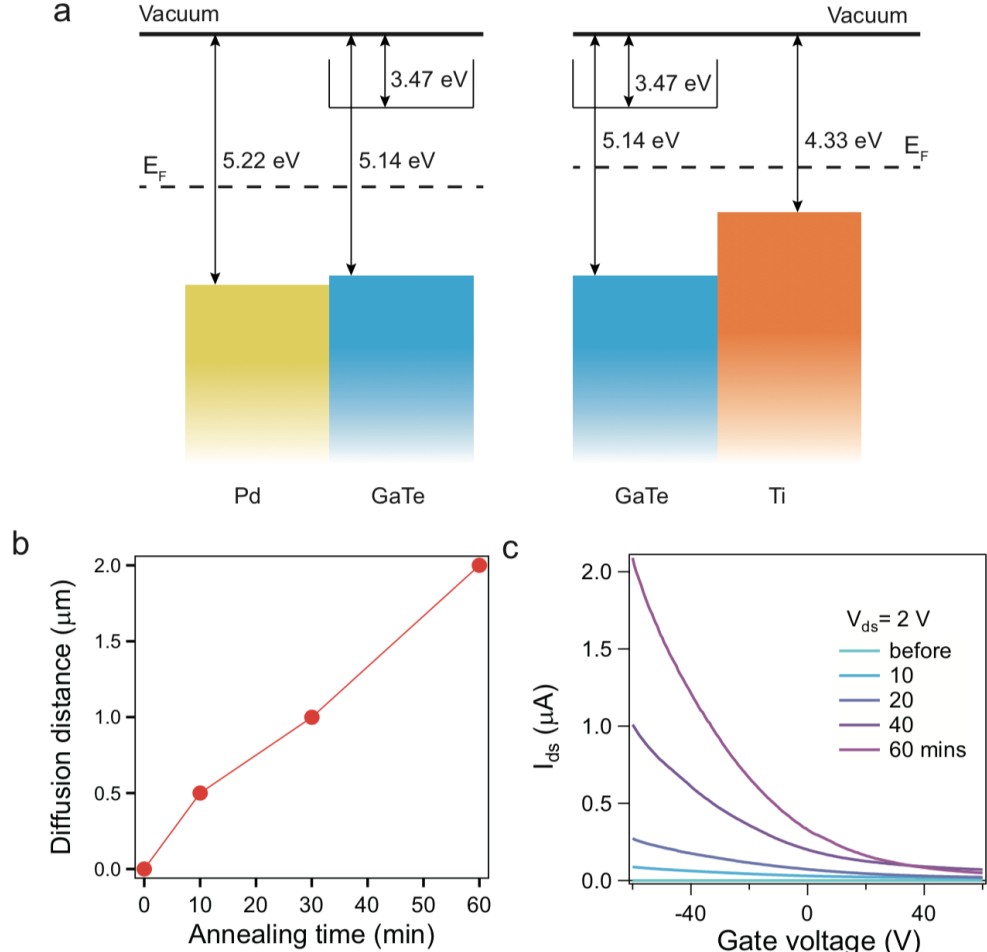

**Figure 2.** (**a**) Schematic of the band alignments of GaTe, Pd, and Ti. Lower contact barrier can be seen between GaTe and Pd than that between GaTe and Ti. (**b**) Diffusion distance of Pd as a function of annealing time in Pd/Ti/Au devices. (**c**) Transfer curves of devices with different annealing time.

After that, we measured the electrical transport performance of the same GaTe device with Pb/Ti/Au electrodes in Figure 2c after 60 min annealing process at 250 K. As shown in Figure 3, the device shows a typical p-type behaviour. In Figure 3a, colour map of source-drain current ($I_{ds}$) versus bias voltage ($V_{ds}$) at various gate voltage ($V_g$) ranging from $-80$ V to 80 V indicates ON and OFF states of FET with the positive and negative $V_{ds}$ at the hole and electron side, respectively. Line cuts of *I-V* along various $V_g$ in Figure 3a are plotted in Figure3b, linear output curves were observed at wide gate voltage range from $-80$ V to 80 V. The $I_{ds}$ reaches up to micro-ampere at the hole side and decreases with increasing gate voltage. Figure 3c plots the line cuts of transfer curves at fixed $V_{ds}$ in Figure 3a. With the variation of $V_{ds}$, our GaTe FET device exhibits typical p-type field effect. Multiple samples confirm the same behaviour, which is in agreement with previously reported results [12]. We will show in the coming parts that the lowered contact barrier was achieved by the controlled micro-alloying method.

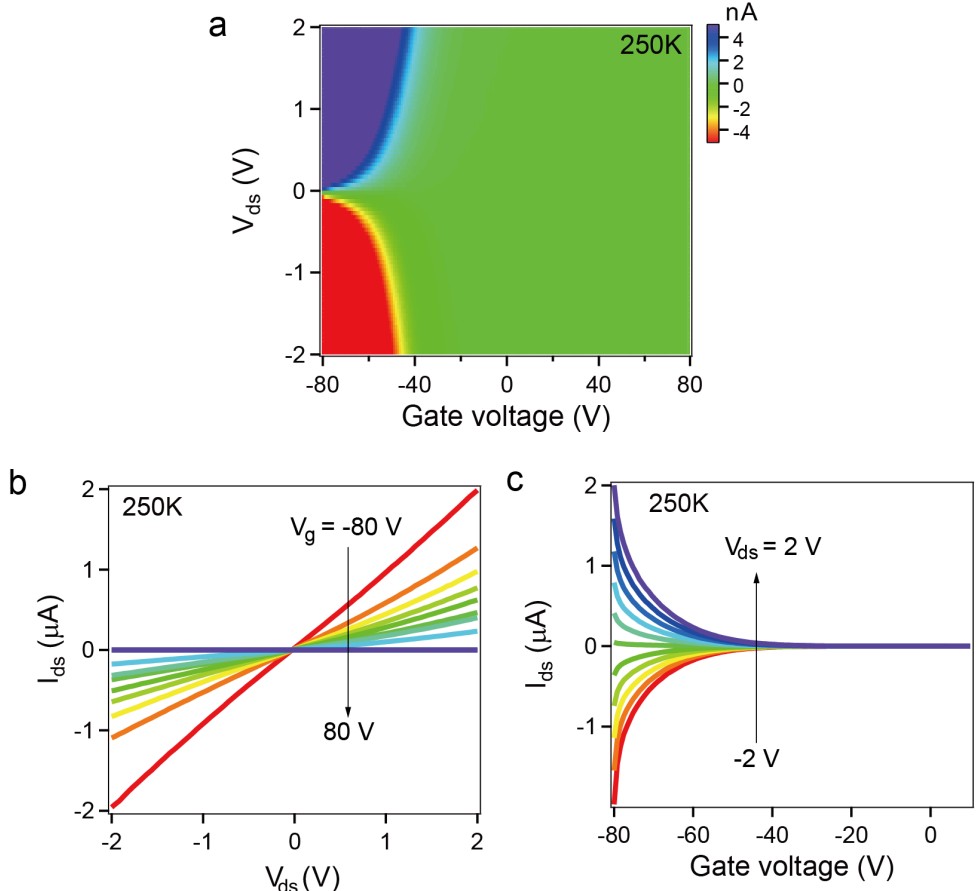

**Figure 3.** (**a**) Colour map of output curves as a function of various gate voltage for a typical GaTe FET with Pd/Ti/Au electrodes. (**b**) Line cuts in a, with *I-V* curves along various gate voltage, linear relationship of current and voltage is observed. (**c**) Transfer curves ($I_{ds}$ vs. $V_g$) along fixed bias voltage.

To further understand the origin of enhanced electrical performance, we carried out the KPFM analysis. Figure 4a shows the optical image of a typical device with Pd/Ti/Au after annealing. Micro-alloying is distinguishable in the electrode and its adjacent area. Figure 4b is the AFM image of the area selected in Figure 4a. Obviously the micro-alloying area is not reflected in the height sensor image or morphology. However, as shown in Figure 4c, an apparent area of gradient potential distribution is observed between the electrode and contacted GaTe. Given the related link between Schottky barrier and work function, the KPFM result confirms the role of barrier lowering played by Pd/GaTe interface. It is suggested in the KPFM image that potential in the contact adjacent area is very close to the value of palladium. To evidence the role played by Pd-alloying on the improvement of potential distribution in the contact area, we carried out EDS mapping at the identical selected area in Figure 4a. EDS mappings in Figure 4d–f confirm the diffusion of Pd in the micro-alloying area with the distinguishable area in the optical image and the KPFM image. Notice that no Au or Ti diffusion/alloying was detected in the adjacent area.

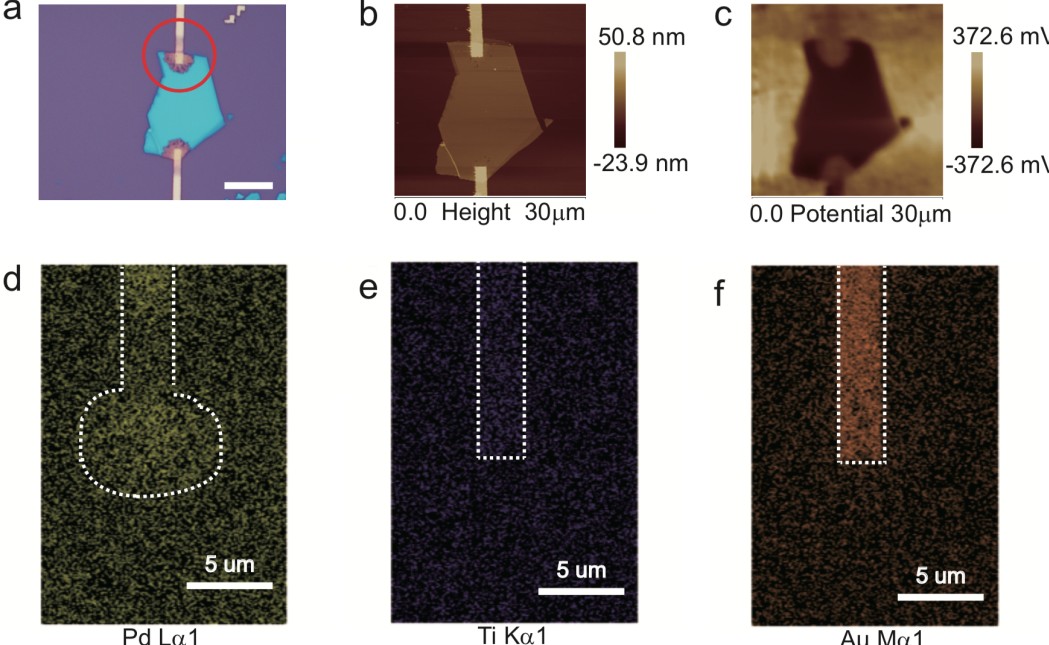

**Figure 4.** (**a**) Optical image of a typical device as fabricated with Pd/Ti/Au electrode; scale bar is 10 μm. (**b**) atomic force microscopy (AFM) image of selected area in (**a**), the thickness of GaTe is 16.6 nm. (**c**) Kelvin probe force microscope (KPFM) image of identical area in (**b**), a gradient distribution of potential is observed. (**d**–**f**) energy dispersion spectrum (EDS) mapping of the different metal electrodes area of the devices. Noticeable diffusion of Pd is observed while Ti and Au maintain their original distribution.

Finally, we carried out low temperature measurement on the as-prepared Pd-alloy-contacted few-layered GaTe devices. A source-drain bias $V_{ds}$ was applied to one of the pairs of metal electrodes as shown in Figure 5a and gate voltage was applied via the bottom doped silicon. Figure 5b depicts the $I_{ds}$ mapping with varied gate voltage at different temperature from 300 K to 4 K, and the following figure is the corresponding $I_{ds}$-$V_g$ curves. A current in micro-ampere was observed from ambient temperature to 4 K. Furthermore, the current on/off ratio reaches $\sim 10^4$ at room temperature. From the data presented in Figure 5b,c, the carrier mobility can be extracted using the expression $\mu = \frac{dI_{ds}}{dV_{bg}} \cdot \frac{L}{WC_{bg}V_{ds}}$ [35], where $L$ is the channel length, $W$ is the channel width, $C_{bg} = \frac{\epsilon_0}{(d_1/\epsilon_1) + (d_2/\epsilon_2)}$ is the capacitance per unit area, and $d_1$ $d_2$ are the thicknesses of $HfO_2$ dielectric and $SiO_2$ dielectric, respectively. As shown in Figure 5d, the mobility is still rather low and decreases with decreasing temperature. Generally, an insulating behaviour dominates in the fabricated devices. Further efforts may be needed to obtain even higher mobility in few-layered GaTe devices.

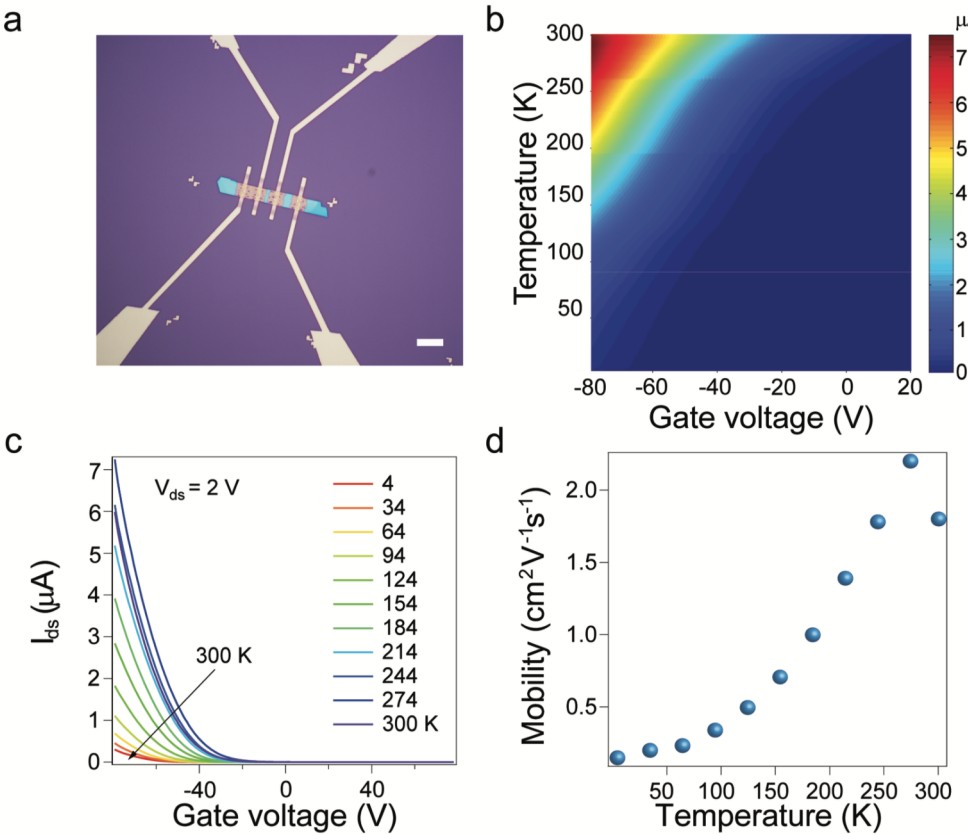

**Figure 5.** (**a**) Optical image of a typical GaTe device; scale bar is 10 μm. (**b**) Colour map of $I_{ds}$-$V_g$ as a function of temperature. (**c**) Extracted $I_{ds}$-$V_g$ at different temperature in (**b**). (**d**) Extracted mobility at different temperature in (**c**).

## 4. Conclusions

In conclusion, we have manifested a controlled micro-alloying method to introduce electrical contacts in few-layered GaTe devices via Pd/Ti/Au electrode and HfO$_2$ substrate. We show that, benefiting from micro-alloying, few-layered GaTe FETs with rather linear *I-V* curves and an on/off ratio of ~$10^4$ can be achieved. Kelvin probe and EDS mapping suggest that the effect of Schottky barrier lowering may come from the diffusion of Palladium into GaTe channel. Our findings provide a new strategy to engineer the electrode contacts for future device applications based on 2DLMs. Moreover, our approach is compatible with traditional semiconductor manufacturing and is feasible and expandable for the broad application of two-dimensional nano devices.

**Author Contributions:** Conceptualization, X.L. and H.W.; methodology, X.X. and X.S.; formal analysis, X.X., X.S. and H.W.; investigation, X.X. and X.S.; data curation, X.X. and X.S.; writing–original draft preparation, X.X., X.S. and H.W.; writing–review and editing, X.X., X.S., H.W. and X.L.; project administration, X.L.; funding acquisition, X.L. All authors have read and agreed to the published version of the manuscript.

**Funding:** This work is supported by the National Key R&D Program of China (2019YFA0307800) and is supported by the National Natural Science Foundation of China (NSFC) with Grant 11974357, U1932151, and 51627801.

**Acknowledgments:** The authors are grateful for the helpful discussions with Baojuan (Donna) Dong.

**Conflicts of Interest:** The authors declare no conflict of interest.

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
