# Peer review of "Introducing Electrode Contact by Controlled Micro-Alloying in Few-Layered GaTe Field Effect Transistors"

_crystals, doi:10.3390/cryst10030144_

Round 1

Reviewer 1 Report

The paper is well written with clearly described methods and presented results. The conclusions may be considered as correct and well supported by the results. There are only a few mistakes in the text which are noted bellow.

Despite of the indisputable effort invested into the research, the Gallium Telluride is not that much promising material as presented by the authors in introduction as there are only 9 papers in 2018 and 8 papers in 2019 related to this material published. Only about 100 papers published so far. This statistics makes the paper very original and novel on one side but not very interesting on the other side. Nevertheless, there is a certain possibility for the paper to be cited and therefore I am recommending to publish it after text corrections.

Corrections:

row 15: "gallium telluride" instead of "telluride gallium"
row 37: ...has an impact...
row 47: Ingredient powders ...
row 53: ... oxygen, and atomic force ...
row 86: ... in Figure 3(a) are plotted in Figure 3(b) ...
Fig. 4 caption: (a) Optical image of a typical ...
Fig. 5 caption: (a) Optical image of a typical ...

Author Response

We thank Reviewer 1 for your positive comments. We have corrected the manuscript correspondingly and the corrections are highlighted in the main text in the revised version.

row 15: "telluride gallium" has been corrected into "gallium telluride".

row 37: ...has a impact... has been corrected into … has an impact …

row 47: Ingredient powers ... has been corrected into Ingredient powders ...

row 53: ... oxygen, atomic force ...has been corrected into ... oxygen, and atomic force ...

row 86: ... in Figure 2(a) are plotted in Figure 3(b) ...  has been corrected into ... in Figure 3(a) are plotted in Figure 3(b) ...

row 86: Line cuts of IV ...  has been corrected into Line cuts of I-V ...

Fig. 4 and Fig. 5caption: (a) Optical of a typical ...  have been corrected into caption: (a) Optical image of a typical ...

We have looked through our paper thoroughly. Except for the minor corrections listed above. Some other minor corrections are also corrected, and all the corrections are highlighted by red color in the main text.

We thank the referee again for your very helpful comments.

Reviewer 2 Report

I believe that the current manuscript may be accepted for publication. Although is not an outstanding paper, it combines theory with experiments and might be of general interest to the public. I would recommend an overall improvement of English usage.

Author Response

We thank Reviewer 2 for your positive comments and valuable suggestions.

We’ve checked our manuscript in detail especially the Results and Discussion sections.

Compared with previous version, we added more details in the experiment part and corrected some contents in the Results and Discussion parts with more precise descriptions. English has been polished as well.

All the corrections are highlighted by red color in the revised main text.

We thank the referee again for your very helpful comments.